# The SLIT/ROBO Pathway in Liver Fibrosis and Cancer

**DOI:** 10.3390/biom13050785

**Published:** 2023-05-01

**Authors:** Sreenivasulu Basha, Brady Jin-Smith, Chunbao Sun, Liya Pi

**Affiliations:** Department of Pathology, Tulane University School of Medicine, 1430 Tulane Ave, New Orleans, LA 70112, USA

**Keywords:** ROBOs, SLITs, liver fibrosis, liver cancer, hepatocellular carcinoma, cholangiocarcinoma

## Abstract

Liver fibrosis is a common outcome of most chronic liver insults/injuries that can develop into an irreversible process of cirrhosis and, eventually, liver cancer. In recent years, there has been significant progress in basic and clinical research on liver cancer, leading to the identification of various signaling pathways involved in tumorigenesis and disease progression. Slit glycoprotein (SLIT)1, SLIT2, and SLIT3 are secreted members of a protein family that accelerate positional interactions between cells and their environment during development. These proteins signal through Roundabout receptor (ROBO) receptors (ROBO1, ROBO2, ROBO3, and ROBO4) to achieve their cellular effects. The SLIT and ROBO signaling pathway acts as a neural targeting factor regulating axon guidance, neuronal migration, and axonal remnants in the nervous system. Recent findings suggest that various tumor cells differ in SLIT/ROBO signaling levels and show varying degrees of expression patterns during tumor angiogenesis, cell invasion, metastasis, and infiltration. Emerging roles of the SLIT and ROBO axon-guidance molecules have been discovered in liver fibrosis and cancer development. Herein, we examined the expression patterns of SLIT and ROBO proteins in normal adult livers and two types of liver cancers: hepatocellular carcinoma and cholangiocarcinoma. This review also summarizes the potential therapeutics of this pathway for anti-fibrosis and anti-cancer drug development.

## 1. Introduction

Liver cancer is the sixth most common form of cancer, accounting for 8.2% of cancer-related deaths [1]. Hepatocellular carcinoma (HCC) and cholangiocarcinoma (CCA) are the most frequently diagnosed types of liver cancer. HCC and CCA mainly result from parenchymal cell damage in chronic liver insults, such as viral hepatitis, alcohol abuse, hemochromatosis, obesity, metabolic syndrome, or genotoxin exposure. Continuous hepatocyte death can trigger and sustain pro-inflammatory and pro-fibrotic responses, causing abnormal repair processes and ultimately contributing to the development of tumorigenesis or cancer. Basic and clinical research on liver cancer has made extensive progress in recent years, and various signaling pathways have been identified in tumorigenesis and progression in liver cancer.

SLIT and ROBO proteins are large proteins involved in various cell signaling pathways, such as axon guidance, cell proliferation, cell motility, and angiogenesis [2,3,4]. These proteins were first identified as secreted proteins in Drosophila [5,6,7]. SLIT and ROBO homologs were later discovered in rats, mice, and humans [8]. The SLIT/ROBO pathway functions in developmental and physiological processes. Abnormal expression of either structural gene can lead to cancer development, progression, and metastasis [9]. This review summarizes the past and current literature to provide an overview of the SLIT/ROBO pathway in liver fibrosis and cancer. Additionally, we examined distribution patterns of human SLITs and ROBOs in normal adult livers, HCC, and CCA based on publicly available data.

## 2. SLIT Ligands and ROBO Receptors in Normal Human Liver Cells

SLIT proteins are highly conserved across species [10]. They are usually secreted for association with cellular membranes and the extracellular matrix (ECM). Most vertebrates have 3 SLIT genes (SLIT1-3), which encode proteins of 200 kDa. As shown in Figure 1, SLIT proteins contain four stretches of leucine-rich repeat (LRR) domains (also known as D1-D4), seven to nine epidermal growth factor (EGF) repeats, an Agrin-Perlecan-Laminin-Slit (ALPS)/Laminin-G-like domain, and a cysteine knot at the C-terminus [11]. Additionally, full-length SLITs can be cleaved into smaller isoforms containing 140-kDa N terminal and 55-kDa C terminal fragments, respectively [4].

The first three ROBO genes have been reported in Drosophila. *Caenorhabditis elegans* has one ROBO ortholog (SAX-3), *chick* and *Xenopus* have three, and mammals and zebrafish have four [12]. ROBO receptors are single-pass transmembrane proteins that lack autocatalytic or enzymatic activity, implying that downstream signaling and scaffolding molecules mediate their function. ROBO1-3 proteins consist of five immunoglobulin-like domains (Ig1-5) and three fibronectin repeats (FNIII 1-3) [13,14,15]. The SLIT1-3 ligands can bind to these ROBO receptors via their LRR2 domain to the first Ig domain (Ig1) [16,17]. In contrast, ROBO4 only has two Ig and one Fn III domain and is mainly found in endothelial cells with a role in angiogenesis [9]. Likewise, the interaction of SLITs to ROBO4 remains elusive, highlighting a comparative lack of understanding.

Transcriptional regulation of *Robo1, Robo2,* and *Robo3* has been comparatively underreported using experimental systems. Yes-associated protein (YAP) signaling for pancreatic progenitor identity controls Robo1 and Robo2 expression in mouse embryos [18]. In Drosophila, Hox transcription factors during neuron migration regulate Robo2 levels [19]. In the spinal cords and brains of mouse embryos, retinoblastoma (Rb) represses Robo3 [20,21]. Unlike the prior, Robo4 transcriptional regulation has been characterized in greater detail. GA-binding protein (GABP) and specificity protein (SP)1 bind to the Robo4 proximal promoter to regulate its basal promoter activity. Activation of the SP1 element at position −1912/−1908 and the nuclear factor kappaB (NF-kB) element at position −2753/−2220 during hyperglycemia and inflammation can enhance Robo4 expression [22,23,24,25,26,27]. Finally, an activator protein-(AP) 1 binding motif for a c-Jun/c-Jun complex and a c-Jun/Fra-1 complex has been found at position –2875 in the mouse Robo4 promoter for regulation in endothelium [24].

The liver is comprised of various types of cells. Hepatocytes supply roughly 80% of liver cells and power essential liver functions, such as metabolism, biosynthesis, and detoxification. Bile ductular epithelial cells (BDECs) are the other type of parenchymal cells in the liver that form bile ducts to carry out bile acid drainage. Vascular endothelial cells (ECs) form vessel walls to maintain blood flow. Hepatic stellate cells (HSCs) are typically vitamin A-storing cells found in the Space of Disse region between the sinusoid and hepatic plates. Kupffer cells are residual macrophage cells in the liver, which situate near blood vessel walls in sinusoids as part of immune surveillance activity [28]. The liver also contains numerous blood cells, including T cells, B cells, and erythroid cells.

To examine expression patterns of *SLITs* and *ROBOs* in human adult livers, we utilized the public database Human Protein Atlas (https://www.proteinatlas.org, accessed on 2 January 2023) and extracted single-cell expression data of all genes. Transcript profiling in this database was based on a combination of 2 transcriptomics datasets (HPA and GTEx) corresponding to 14,590 samples from 54 different human normal tissue types [29]. Represented in Figure 2A, B, the *ROBO1* transcript was found in seven cell types (hepatocytes, vascular EC, fibroblasts, Kupffer cells, smooth muscle vessel cells, B cells, and macrophages) based on the maximal number of transcripts per million (nTPM). *ROBO2* was observed in five cell types (hepatocytes, vascular EC, fibroblasts, smooth muscle vessel cells, and macrophages). *ROBO3* was expressed in six cell types (fibroblasts, smooth muscle vessel cells, T cells, plasma cells, B cells, and macrophages). Lastly, *ROBO4* was seen in seven cell types (hepatocytes, vascular EC, fibroblasts, smooth muscle vessel cells, T cells, plasma cells, B cells, and macrophages). Likewise, the expression patterns of the three *SLIT* ligands varied by cell type. *SLIT1* was found in five cell types (hepatocytes, Kupffer cells, T cells, plasma cells, and macrophages). *SLIT2* was in seven cell types (hepatocytes, vascular EC, fibroblast cells, Kupffer cells, smooth muscle vessel cells, BDEC, and macrophages). Expressed in only two cell types (vascular EC and smooth muscle cells), *SLIT3* was the most specific of either family. Notably, vascular EC, fibroblasts, and smooth muscle cells were the primary cellular sources for the *ROBO* and *SLIT* genes, except *SLIT1* and *ROBO3*. Nevertheless, low levels of these genes were also detectable in other cell types (B cells, T cells, BDEC, macrophage, and Kupffer cells) within adult human livers. These differential expression profiles indicate that mesenchymal components (EC, fibroblast cells, smooth muscle cells) utilize the majority pools of *SLIT* and *ROBO* genes, while the other cell types may employ them for unique needs during normal human liver homeostasis.

## 3. SLIT/ROBO Pathway in Liver Fibrosis

Fibrosis is a wound-healing reaction that causes tissue scarring by producing and depositing extracellular matrix (ECM) proteins, such as collagen fibers [30,31]. The liver can self-regenerate under acute injury; however, hepatic self-repair can be insufficient during chronic liver injury, resulting in persistent inflammatory and fibrotic responses. Therefore, various etiologies, including hepatitis B and C infection, alcohol misuse, non-alcoholic steatohepatitis (NASH), cholestasis, and autoimmune hepatitis, can cause liver fibrosis. Progressive hepatic fibrosis leads to cirrhosis, in which liver cells cannot function effectively due to reduced blood flow to the liver and the production of fibrous scars and regenerating nodules [32]. Liver damage during hepatic fibrogenesis can transform quiescent HSC into an activated form responsible for accumulating ECM proteins [33]. Furthermore, liver damage induces profibrotic cytokines and growth factors, which activate HSCs to produce α-smooth muscle actin (α-SMA) and ECM, leading to liver fibrosis [34,35,36].

One of the key causal factors in alcoholic liver disease (ALD) is excessive alcohol consumption [37]. Because ALD can develop into cirrhosis, alcoholic liver fibrosis (ALF) is considered a turning point in the disease progression [37]. In current studies on ALD, oxidative stress, defined as an imbalance of oxidation and antioxidation, is receiving more attention. The formation and accumulation of reactive oxygen species (ROS) in the liver are caused by ethanol oxidation by the microsomal oxidizing enzyme cytochrome P450 2E1 (CYP2E1) and alcohol dehydrogenase. Alcohol generates free radicals and reactive oxygen species, which oxidize cellular membranes, causing lipid peroxidation and oxidative stress [38]. The main enzyme responsible for metabolizing a range of low-molecular-weight compounds in human livers is CYP2E1, which also happens to be the most prevalent CYP isoform. Some of the compounds that CYP2E1 metabolizes include ethanol, acetaminophen, benzene, and carbon tetrachloride. Additionally, it plays a crucial role in metabolizing cancer suspects, such as nitrosamines and azo compounds [39,40].

The fibrotic activity of the SLIT/ROBO pathway occurs through the activation of downstream signaling, including enhanced phosphorylation of PI3/Akt molecules in hepatic stellate cells. This effect may differ from the numerous Cytochrome P450 (CYP) enzyme pathways, a highly expressed group of hemeproteins that catalyze the metabolism of both exogenous and endogenous xenobiotics in the liver. The impaired function of CYP enzymes can lead to the development of liver diseases [41]. Previous reports have shown that CYP2C and CYP2D are the most affected isoform CYP enzymes in healthy and liver disease individuals like ALD and cirrhosis [42]. CYP enzymes are involved in polyunsaturated fatty acids (PUFA) metabolism and are among the most enriched genes in HCC [43]. CYP2A6 expression was closely correlated with tumor grades and receiving a favorable prognosis in a distinct cohort [44]. Notably, the prognostic panel of genes showed that CYP26A1, CYP2C9, and CYP4F2 are risk markers for ICC and HCC [43]. Consequently, several CYP4 enzymes are associated with favorable outcomes in HCC [45]. CYP metabolites have also been associated with the liver cancer diagnostic marker alpha-fetoprotein (AFP) [46,47]. Whether different CYP450 enzymes mediate these pathological actions via crosstalk with SLIT/ROBO signaling remains to be determined.

Slit/Robo pathways can be activated in a variety of clinical and experimental conditions post-liver damage. In carbon tetrachloride (CCl_4_)-damaged mice, Slit2-Robo1 signaling promotes liver fibrosis by activating HSCs [48]. Subsequent studies have demonstrated that transcriptomic analysis reveals significant upregulation of axon guidance signaling pathways in experimental diethylnitrosamine (DEN) and thioacetamide (TAA)-induced liver fibrosis. Additionally, the upregulation of genes encoding the ligand Slit2 and membrane receptor Robo2 within this pathway has been confirmed in TAA-induced fibrotic liver [49]. Coll et al. have used 3,5-diethoxycarbonyl-1,4-dihydrocollidine (DDC) to trigger the ductular reaction (DR) that secretes Slit2 and activates Robo1 on vascular EC, leading to angiogenesis after liver injury [50]. Furthermore, serum levels and hepatic expression of SLIT2 are significantly elevated in patients with primary biliary cirrhosis (PBC) [51]. Slit2 overexpression in transgenic mice is more susceptible to fibrosis and injury induced by bile duct ligation, while its deletion reduces cholestatic fibrosis in mice by HSC activation [51]. Slit2 is also identified as liver immune microenvironment-related hub genes in livers of biliary atresia [52].

We have recently reported mouse models suggesting that deletion of the profibrotic connective tissue growth factor (*Ctgf*) gene significantly reduces expression of fibrosis-related genes, such as Slit2, aSMA, and Collagen type I, during CCl_4_-induced liver fibrosis in mice [53]. Alternatively, ectopic expression of the Ctgf protein increases Slit2, promotes HSC activation, and potentiates fibrotic responses after CCl_4_ intoxication. As shown in Figure 1, N terminal ROBO receptors carrying the first two Ig domains can be used to make FC chimeric fusion proteins that sequester Slit2 from presentation to Robo receptors and inhibit corresponding biological processes [54,55,56]. Soluble ROBO1N-Fc chimera protein can inhibit PI3K and Akt activation in Ctgf and Slit2-stimulated primary murine HSC [53]. These findings collectively suggest that Slit2/Robo1 signaling mediates HSC activation and contributes to the pathogenesis of liver fibrosis through crosstalk with Ctgf.

## 4. The SLIT/ROBO Pathway in HCC

HCC accounts for nearly 90% of liver cancer cases. According to the Surveillance Epidemiology End Results (SEER), HCC has been the most rapidly rising cause of cancer-related deaths in the United States since the early 2000s. Epidemiological studies have identified several risk factors for HCC, including infection with hepatitis B and C viruses, exposure to chemicals, excessive alcohol consumption, obesity, and diabetes. The primary connection between cancer and SLIT/ROBO signaling was initially shown in 1995 [57]. Further studies have demonstrated the absence of exon 2 in *ROBO1* within cell lines of both lung and breast cancer [58]. Numerous studies also implicate that hyper-methylation (epigenetic inactivation) of the promoters for *SLIT1-3* and *ROBO1-4* occur in various cancer types [59,60,61,62,63]. Abnormal expression patterns of *SLIT-ROBO* genes have also been demonstrated in numerous cancer types. Overexpression of *ROBO1* has been shown in breast carcinoma tissue samples, and *SLIT2* stimulates migration in breast cancer cell lines [64]. The research conducted by *Ito* et al. [30] has revealed that *SLIT1* and *SLIT3* genes are upregulated in prostate tumors, but are only slightly expressed, along with *SLIT2*, in poorly differentiated HCC [65]. Human melanoma and breast cancer cells expressing CXCR4, ROBO1, and ROBO2 are activated to migrate via CXCL12 [66]. In contrast, the interaction of SLIT2-ROBO inhibits breast cancer cell chemotaxis, chemo invasion, and adhesion [66].

Multiple microRNAs have been shown to control Slit2 and Robo1 post-transcription in tumor cells. Overexpressed miR-29-3p can prevent HCC cell proliferation, migration, invasion, and metastasis [67]. MiR-588 expression is significantly higher in hypoxic glioma cells compared to normoxic glioma cells. In glioma, ROBO1 is a direct and functionally important target of miR-588. ROBO1 knockdown inhibited glioma invasive, migratory, and VM-formation abilities by suppressing the expression of matrix metallopeptidase 2 (MMP2) and matrix metallopeptidase 9 (MMP9) [68]. Links between *ROBO1* upregulation and poor prognosis, immune cell enrichment, and cell proliferation have been found in HCC. Overexpressed ROBO1 in HCC cells can counteract suppressed cell proliferation and enhanced cell apoptosis enacted by miR-152-3p mimics. The miR-152-3p/ROBO1 signaling axis promotes cancer progression and offers a potential immunotherapeutic target for HCC [69]. MiR-490-5p expression is lower, whereas ROBO1 expression is higher in HCC tissues and cells. MiR-490-5p can reduce cell growth, migration, and invasion in Hep3B cells, but enhance apoptosis by inhibiting ROBO1 function during HCC cell proliferation, migration, and invasion [70].

Increasing evidence of SLIT-ROBO deregulation has been found in primary HCC. Some have proposed ROBO1 as a novel HCC antigen and a therapeutic and diagnostic target [65]. While HCC induces ROBO1, its expression is limited in normal tissues. Additionally, the ectodomain of ROBO1 has been detected in the sera of HCC patients [65]. Quantifying *SLIT-ROBO* transcripts in HCC cell lines, normal liver tissues, and tumor liver tissues suggests they become upregulated during liver cancer development [71]. In a recent study, intense expression of SLIT2 has been found in untreated HepG2 cells [34], indicating an oncogenic role of this ligand in HCC. Moreover, SLIT2 and ROBO1 have also been reported to inhibit tumor metastasis by reducing cell adhesion and increasing cell migration and metastasis. *SLIT2* silencing and *ROBO1* overexpression in Sk-hep-1 cells are associated with decreased cell adhesion and increased cell migration and metastasis [72]. Furthermore, SLIT2 and ROBO1 seem to mediate a novel function in inhibiting epithelial cell motility and invasion mediated by hepatocyte growth factor (HGF)/MET receptor [73].

We detected an 8.6-fold increase of *ROBO1* and a 1.99-fold increase of *SLIT2* based on extracted TCGA data shown in Table 1. High levels of the *ROBO1* gene were significantly associated with poor patient survival rates (*p* = 0.0067). Although *ROBO4* has been shown to be specific to vasculature [74], we also observed its 1.35-fold elevation in HCC. Low levels of the *ROBO1* gene are significantly associated with poor patient survival rates (*p* = 0.02). Consistent with our observations, varying levels of the *ROBO4* gene have been observed in HCC cell lines. The existence of side population cells with stem cell characteristics that express vascular endothelial markers partially explains the presence of *ROBO4* transcripts in HCC cells [75]. *ROBO4* is found in poorly differentiated tumors, but its function is not essential for tumor maintenance at this stage of hepatocarcinogenesis [65,76]. In addition, a 2.6-fold upregulation of *ROBO3*, a 1.2-fold upregulation of *SLIT1*, and 1.99-fold upregulation of *SLIT3* were all detected in primary human HCC compared to normal healthy livers. In contrast, primary human HCC displayed downregulation of the *ROBO2* gene (Table 1). Coincidentally, we observed enhancement of several key fibrosis regulators in HCC patients compared to normal controls, including transforming growth factor beta (*TGFB*)*1* (1.18-fold increase), platelet-derived growth factor (*PDGF*)*A* (5.02-fold increase), and *PDGFB* (2.019-fold increase). Moreover, high levels of *TGFB1* were significantly associated with poor prognosis in HCC patients. These differential levels of *SLITs* and *ROBOs* implicated their different contributions to HCC. Future studies are warranted to characterize the functions of these genes in HCC development and to determine whether these genes can be considered diagnostic and prognostic markers, as well as potential anti-HCC drug targets.

## 5. The SLIT/ROBO Pathway in CCA

CCA is a type of cancer arising from the biliary duct, an origin associated with poor prognosis [77], accounting for 10–15% of all primary liver malignancies. CCA incidence and mortality have steadily increased over the past decade. Most CCAs are diagnosed at an advanced stage when treatment options are severely limited. CCA can arise from any biliary tract; therefore, it is classified into intrahepatic, perihilar, and distal CCA, depending on its anatomical origin [78]. So far, two studies have investigated the role of SLIT and ROBO proteins in intrahepatic cholangiocarcinoma (ICC). Low levels of *SLIT2* and *ROBO1* genes are associated with cell proliferation and migration in ICC, and low *ROBO1* is a prognostic factor in ICC patients, implicating the role of this receptor in the progression of the disease [79]. Recently, SLIT2 has been reported as a driver of ICC dissemination and inflammatory cell infiltration [80]. To further understand the expression of *SLITs* and *ROBOs* in CCA, we searched the publicly available Gene Expression Omnibus (GEO) dataset (https://www.ncbi.nlm.nih.gov/geo/query/acc.cgi?acc=GSE26566, accessed on 2 January 2023) that includes 104 freshly frozen CCA tumor samples and 59 matched non-cancerous livers obtained from Australia, Europe, and the United States [81]. As shown in Figure 3, we detected a 2.12-fold downregulation of *ROBO1* (*p* = 0.0194) and a 1.23-fold decrease of *ROBO4* (*p* = 0.0087). In contrast, we observed a 2.2-fold increase of *SLIT2* (*p* = 0.0004), a 2.37-fold increase of *SLIT3* (*p* = 0.025), and 1.259-fold elevation of *ROBO3* (*p* = 0.0422). These data are consistent with previous reports [79,80] and indicate the potential negative regulation of the SLIT/ROBO pathway in CCA development.

## 6. Targeting the SLIT/ROBO Pathway for Anti-Fibrosis and Anti-Cancer Therapy

Potential therapeutic approaches can be applied to target the SLIT/ROBO pathway for anti-fibrosis and anti-HCC treatment. As shown in Figure 1, one approach may utilize ROBO-Fc chimeric proteins. This is achieved by soluble chimeric receptors, in which the extracellular domains of ROBOs are fused to the Fc region of an immunoglobulin, thereby blocking Slit functions and identifying the SLIT binding domains. For instance, a recombinant protein consisting of only the first two Ig domains of ROBO1, which allows it to permeate better in tissue cultures, can inhibit SLIT protein function [82]. Numerous studies have demonstrated successful applications of Robo-Fc chimeric proteins in inhibiting exogenous and endogenous Slit functioning [53,83,84,85]. In addition, a series of microRNAs (miR-152-3p, miR-146a-5, miR-588, miR-218, circular RNA_0001495, miR-548a, miR-32, etc.) can regulate levels of ROBO1 [68,69,86,87,88,89]. RNA interference can be applied to block the SLIT/ROBO signaling during liver fibrosis and cancer. For example, miR-490-5P and miR-152-3P act as tumor suppressors and can be used to target ROBO1 in hepatic tumorigenesis [70]. Lastly, ROBOs as surface antigen represents a novel immunotherapeutic target and sensitive serologic marker for HCC. Indeed, ROBO1 has been considered a potential target antigen for radioimmunotherapy in human HCC [65]. A recent breakthrough using the immune checkpoint inhibitors (ICIs) targeting programmed cell death ligand 1 (PDL1) and vascular endothelial factor (VEGF)A indicates the longer overall survival of unresectable cases in comparison to the use of sorafenib in advanced HCC patients [90]. Blocking ICIs has changed the landscape of liver cancer therapy. ***ROBO1*** in clinical HCC samples is significantly and positively correlated with multiple ICIs, including cytotoxic T-lymphocyte-associated protein 4 (CTLA-4), Programmed cell death protein 1 (PD-1), CD274, T-cell immunoreceptor with Ig and ITIM domains (TIGIT), and butyrophilin subfamily 2 member A1 (BTN2A1) and BTN2A2 [69]. ROBO1, together with multiple ICMs, including CTLA-4, PD-1, and programmed death-ligand 1 (PD-L1), have been considered potential target antigens for effective HCC radioimmunotherapy [91]. This receptor has also been proposed as a candidate for peptide-based vaccines in a larger antigenic repertoire for upcoming clinical anti-HCC studies [92]. Administering monoclonal antibodies (Mabs) against ROBO1 has shown anti-cancer potentials [93]. Application of the 90Y-anti-ROBO1 Mab on HCC xenograft tumors in nude mice significantly suppresses tumor growth without necrosis or fibrosis [94]. All the studies show the possibilities for anti-cancer therapeutics; however, some side effects may occur when SLITs/ROBOs are targeted in vivo. For example, aberrant lung development is observed because of the genetic removal of Ig1 and Ig2 [95]. Further characterizations of SLIT/ROBO signaling in liver fibrosis and HCC are necessary to identify therapeutic potentials and avoid pitfalls.

## 7. Conclusions

The SLIT/ROBO pathway plays crucial roles in organ development, homeostatic maintenance, and tumorigenesis. Apart from its role in cell migration, this pathway regulates proliferation, apoptosis, adhesion, and angiogenesis in normal and tumor cells. Moreover, SLIT2 can act on ROBO1 and ROBO2 receptors and promote liver fibrosis by activating downstream PI3K/Akt signaling. Additionally, our analyses of Human Protein Atlas and TCGA datasets detected differential distribution of SLITs and ROBOs in different liver cell types. These proteins may exert a context-dependent function in a cell-specific manner. For example, *ROBO1* upregulation in HCC was correlated with poor survival rates in patients, whereas this receptor was downregulated in CCA. These opposite patterns of *ROBO1* implicate a pro-tumorigenic role in HCC and a tumor-suppressive function in CCA. A better understanding of the action mode of the SLIT/ROBO pathway in liver fibrosis and cancer requires the development of genetic animal models, such as conditional knockouts of *ROBOs* and *SLITs*. These tools are important for the study of the contributions of SLIT/ROBO signaling to fibrotic microenvironments in hepatic parenchyma. This knowledge is fundamental to any therapeutic development for treating liver diseases and liver cancer.

## Figures and Tables

**Figure 1 biomolecules-13-00785-f001:**
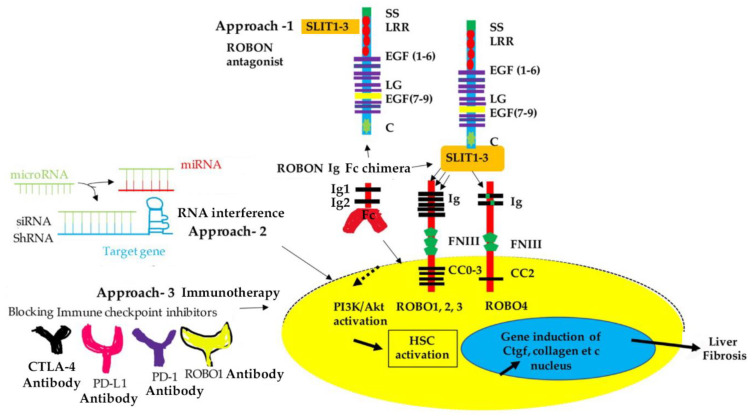
SLIT/ROBO signaling. SLIT1-3 ligands can bind to ROBO1-3 in cell type-specific manners. Within activated HSC, SLIT2 binding to ROBO1 or ROBO2 has shown to activate downstream PI3K/Akt signaling, leading to the induction of pro-fibrotic factors, such as Ctgf and collagen. Three potential approaches that target the Slit/Robo pathway are also summarized. Approach 1 is to use Soluble RoboN Fc chimera to sequester the Slit ligand. Approach 2 is to take advantage of the RNA interference technique (microRNA or siRNA). Approach 3 is based on ROBO1 surface antigen as an immunotherapeutic target. ROBO1 may be used with immune checkpoint inhibitors, including CTLA-4, PD-L, and PD-1, for peptide-based vaccines in anti-HCC treatment. Domain structures of the vertebrates SLIT and ROBO proteins are shown. Abbreviations: SS: Signal peptide; LRR: leucine-rich repeat; EGF: epidermal growth factor; LG: laminin G; EGF: epidermal growth factor; C: cysteine knot; IgG: immunoglobulin-l; FN3: fibronectin type III domain; CC0-3: conserved cytoplasmic motif 0-3; CTLA-4: cytotoxic T-lymphocyte-associated protein 4; PD-L1: Programmed death-ligand 1; PD-1: Programmed death protein 1; miRNA: MicroRNA.

**Figure 2 biomolecules-13-00785-f002:**
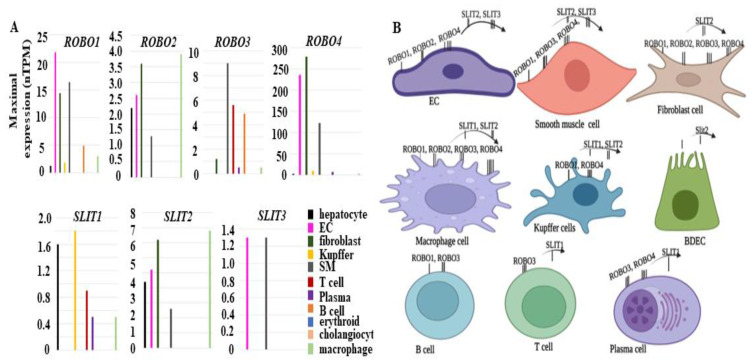
Expression patterns of *ROBOs* and *SLITs* transcripts in different cell types of normal human livers. (**A**) Graphs were generated based on maximal nTPM in Single Cell data of Human Protein Atlas (https://www.proteinatlas.org, 2 January 2023). (**B**) A diagram summarizes *SLIT* and *ROBO* gene product expression patterns in different liver cell types.

**Figure 3 biomolecules-13-00785-f003:**
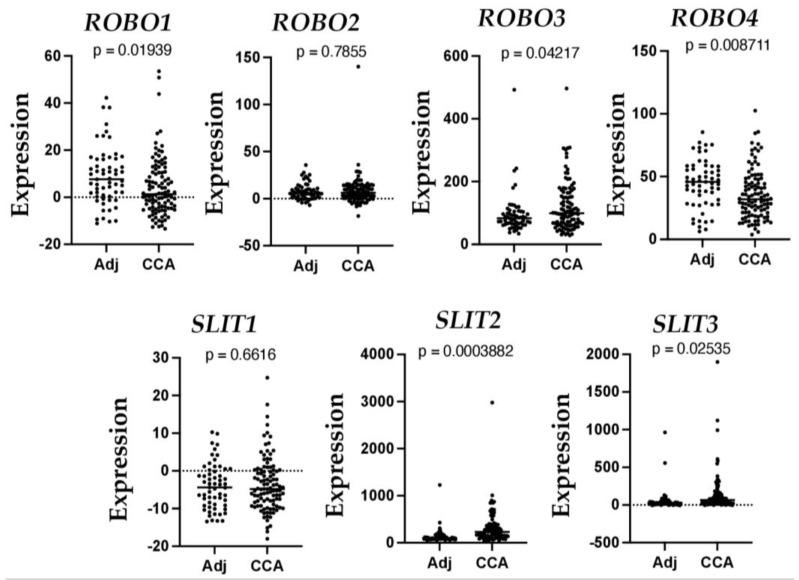
Altered mRNA levels of *SLIT* and *ROBO* in human CCA. Graph data were extracted from a GEO dataset (https://www.ncbi.nlm.nih.gov/geo/query/acc.cgi?acc=GSE26566, accessed on 2 January 2023).

**Table 1 biomolecules-13-00785-t001:** Summary of SLIT/ROBO, TGFB1 and PDGF expression in human primary HCC.

No. of Gene	Gene Name	Normal (N=50) Medium	Median Value Primary HCC (N=371) Medium	Statistical Significance (Normal) Versus Primary HCC)	Statistical Significance For Slit/Robo Expression with Poor Prognosis in Survival Rates
1	ROBO1	0.995	8.545	Up, *p* = 1.62 × 10^−12^	*p* = 0.0067
2	ROBO2	0.047	0.022	Down, *p* = 5.0 × 10^−4^	*p* = 0.8
3	ROBO3	0.741	1.934	Up, *p* < 1 × 10^−12^	*p* = 0.013
4	ROBO4	4.03	5.465	Up, *p* = 1.42 × 10^−8^	*p* = 0.02
5	SLIT1	0.017	0.021	Up, *p* = 2.3 × 10^−3^	*p* = 0.71
6	SLIT2	0.139	0.068	*p* = 8.71 × 10^−7^	*p* = 0.73
7	SLIT3	0.37	0.738	Up, *p* = 1.30 × 10^−7^	*p* = 0.57
8	TGFB1	8.946	10.609	Up, *p* = 2.18 × 10^−11^	*p* = 0.042
9	PDGFA	1.501	7.755	Up, *p* ≤ 1 × 10^−12^	*p* = 0.62
10	PDGFB	2.102	4.246	Up, *p* ≤ 1 × 10^−12^	*p* =0.96

Data are from TCGA database (http://ualcan.path.uab.edu/analysis.html, accessed on 2 January 2023). *SLIT/ROBO* genes with statistical significance (*p* < 0.05) are in bold.

## Data Availability

The data is available on reasonable request.

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
