# Peer review of "The SLIT/ROBO Pathway in Liver Fibrosis and Cancer"

_biomolecules, 2023, doi:10.3390/biom13050785_

Round 1

Reviewer 1 Report

This is a well-written review from Sreenivasulu B. et at., representing a novel field/topic to explore and potentially impactful. There are several points to address:

1. Line 103 -104: It is arbitrary to say “cannot regenerate”; maybe “ineffectively” or “insufficiently” is better?

2. Is there a correlation between SLIM/ROBO with TGFB1 and HGF and PDGF in terms of their expression data from TCGA, HPA, and GTEx? The authors may want to look at how SLIM/ROBO are induced in liver fibrosis, since they show upregulation. And include the correlation data in a revised version to improve.

3. Are there any ncRNAs regulating SLIM/ROBO pathway?

4. What are the known/speculative transcription factors for SLIM/ROBO, which might be included.

5. Are SLIM secreted in an active form or latent form?

6. Are any genetically modified mice of SLIM/ROBO and related studies available? 

Author Response

RE:A point by point response to reviewers comments.

We are grateful for the helpful comments.

Reviewer 2 Report

The current review discusses the roles of the SLIT/ROBO pathway in liver fibrosis and cancer. I have major concerns regarding this review:

1- The most important concern regarding this manuscript is that the authors submitted this manuscript as a review article but at the same time they performed some analysis on online databases. According to my knowledge, the main aim of the review articles is to collect the recent findings from the literature on a specific topic and present that in a review for readers. On the other hand, the authors of the current review did analysis by themselves at several parts in the review (lines 70:99, lines 166:188, and lines 200:213). For sure researchers can analyze the data on the different databases and present their findings through the bioinformatics analysis only or integrate that with experimental findings and finally generate a research manuscript but not a review.

2- While the authors did some computational analysis by themselves, they failed in presenting the data from the literature in an appropriate way. For example, SLIT/ROBO pathway is the main target of this review; nevertheless, the name of the pathway was not reported consistently throughout the whole manuscript. Sometimes SLIT/ROBO and other times Slit/Robo. Line 125 starts with (our mouse models) and according to my knowledge, the authors did not perform experiments in this review. Moreover, some sentences were poorly written (ex: lines 116-117 and line 24).

3- Authors mentioned in the abstract that (This review also summarized the potential therapeutics of this pathway for anti-fibrosis and 12 anti-cancer drug development). I cannot find the summary for that point in the whole manuscript.

4- From lines 62:70, the authors mentioned several points without even a single citation of a reference.

Author Response

RE: A-point by point response to reviewers comments.

We are grateful for the helpful comments  from the editors and reviewers and have carefully revised our manuscript accordingly .

Reviewer 3 Report

In this concise and comprehensive review, authors have summarized the mechanistic contribution of the SLIT/ROBO pathways to liver fibrogenesis as well as hepatic carcinoma and cholangiocarcinoma. Some comments for improvement are suggested below.

MAJOR

ü  Lines 51-52: “…ROBO4 only has two Ig and one FNIII…” This is not clear on figure 1 where ROBO4 has 2 extracellular Ig (correct) and 2 FNII (please check and clarify if this is correct and not contradictory/confusing).

ü  Lines 224-225: It’d be interesting to understand the challenges posed the differential expression of ROBO1 on therapeutics especially targeting this protein in terms of side effects as ROBO1 is essential for instance in axon guidance, cell adhesion etc.

MINOR

·         Line 7: “Liver fibrosis is a common outcome of nearly all types of chronic liver insults/injuries.”

·         Line 24: “…signalling leading to abnormal repair and consequently, tumorigenesis…”

·         Line 26: “…various signalling pathways have been implicated in tumorigenesis…”

·         Lines 35-36: “…based on publicly available data…”

·         Line 97: ….normal human liver…

·         Line 98: Remove “cartoon shows…” to have “…expression patterns of SLIT and ROBO gene products in different cell types…”

·         Line 125: “…Investigation using mouse models suggest that deletion of the Ctgf gene significantly…” As this work is not presenting your previous work in any details.

·         Line 206: Do you mean “2.37-fold reduction” ? Please clarify.

·         Line 216: “…Apart from its ROLE in cell migration…”

·         Line 218: ROBP2? Please correct.

·         Line 221: .”…context-dependent function…”

Author Response

RE:A point -by- point response to reviewers comments.

We. are grateful for the helpful comments from the editors and reviewers and have carefully revised manuscript accordingly.

Reviewer 4 Report

Dear Authors:

The manuscript entitled "The SLIT/ROBO pathway in liver fibrosis and cancer", written by Basha et al. summarizes the potential therapeutics of SLIT/ROBO pathway for anti-fibrosis and anti-cancer drug development. This is an interesting article, but the underlying problem is not described clearly enough.

1. If the authors consider  SLIT/ROBO to be the main "pathway" against in liver fibrosis and cancer, it would be useful to provide the main differences with another pathways such i.e. CYP450.

2. To justify the main objective of this review, it would be useful to describe the mechanism of action of drugs in SLIT/ROBO pathway. I would recommend to add a scheme.

3. Abstract should be improved.

4. Plagiarism was found in the main body.

Kind regards,

Author Response

RE: A point by point response to reviewer’s comments.

We are grateful for the helpful comments from the editors and reviewers and have carefully revised our manuscript accordingly. A point by point response to the reviewers comments, repeated in italic, is given below.

Round 2

Reviewer 2 Report

The authors have not addressed my comments so my final decision is to reject this manuscript.

Author Response

RE: A point-by-point response to reviewers’ comments
We are grateful for the comments provided by the Editors and Reviewers and have carefully revised our manuscript accordingly. A point-by-point response to comments from the Reviewer 2 and the Reviewer 4, repeated in italics, is given below

Reviewer 4 Report

I do not have problem in suggest it to publication

Author Response

(The authors gave the same response as above.)
